# Non-Face-to-Face P2P (Peer-to-Peer) Real-Time Token Payment Blockchain System

**Hyug-Jun Ko [1], Seong-Soo Han [2],* and Chang-Sung Jeong [3],***

1 Visual Information Processing, Korea University, Seoul 12841, Republic of Korea; doltwo@korea.ac.kr
2 Division of Liberal Studies, Kangwon National University, Samcheok 25930, Republic of Korea
3 Department of Electrical Engineering, Korea University, Seoul 12841, Republic of Korea
* Correspondence: sshan1@kangwon.ac.kr (S.-S.H.); csjeong@korea.ac.kr (C.-S.J.);
   Tel.: +82-10-2274-0155 (S.-S.H.)

**Abstract:** With the increase in intelligent voice phishing and the increasing reliance on open banking systems, there has been a rise in cases where individuals' personal information has been exposed, resulting in significant financial losses for the victims. Non-face-to-face transactions in the financial sector face challenges such as customer identification, ensuring transaction integrity and preventing transaction rejection. Blockchain-based distributed ledgers have been proposed as a solution but their adoption is limited due to the difficulty of managing private keys and the burden of gas fees management. This paper proposes a non-face-to-face P2P real-time token payment system that minimizes the risk of key loss by storing private keys in a keystore file and database through a server-based key management module. The proposed system simplifies token creation and management through a server-based token management module and implements an automatic gas-charging function for smooth token transactions. Transaction integrity and non-repudiation are ensured through a transaction confirmation module that uses transaction IDs without exposing personal information. Furthermore, advanced security measures such as blocking foreign IP access and DDoS defense are implemented to securely protect user data. The proposed system aims to provide a convenient, secure and accessible online payment solution to the public by implementing a self-authentication function using a web application that is not limited to smartphones or application platforms.

**Keywords:** blockchain; Symverse; Ethereum; payment; gas; voice fishing; fintech



## 1. Introduction

The rapid advancement of technology and widespread adoption of digital financial services have opened up a new era of convenience and efficiency in transactions. However, along with these benefits, the possibility of financial fraud has also increased and the risk of financial fraud is increasing as the sophistication of voice phishing and the dependence on open banking systems increase [1]. Since these kinds of systems integrate information from various financial institutions, the criminals create potential opportunities to exploit vulnerabilities that may result in significant financial losses to individuals whose personal information is inadvertently exposed [2].

In the area of non-face-to-face transactions, challenges such as customer identity verification, ensuring transaction integrity and preventing transaction refusal remain. These obstacles hinder the introduction of safe and smooth financial services that protect users from fraud while maintaining ease of use. Blockchain technology, with its unique characteristics of decentralization, immutability and transparency, has been proposed as a potential solution to resolve these challenges. Fintech technology using blockchain can be used as a means of small payments at lower fees than bank transactions on a platform based on trust, as an automatic payment method to respond to recurring payments, as a safe transaction service to prevent fraudulent transactions in second-hand transactions,

including bank inspection hours, and to respond to bankbook blackmail, a new type of voice phishing that sends random transmissions to bankbook account numbers exposed to online shopping malls.

A coin in a blockchain is a value object for transactions created by a blockchain mainnet project and transmitted on the mainnet and is used as a fee to record them. Since transactions use both value and cost, it is difficult to transfer full value, and since value fluctuates across exchanges, it is difficult to apply in real life. Blockchain mainnets are enabling the creation of tokens for various purposes to enable fintech, with the Ethereum mainnet providing a way to create ERC-20 tokens and ERC-721 tokens for NFTs [3]. The token is used as a proof of value and Ethereum can be used as the gas that powers smart contracts (a fee for the transfer) to transfer the full value through a wallet (a wallet is a set of programs that store your address and private key on the blockchain and sign transactions as they occur). Therefore, each wallet you use requires gas, which is used as a fee, to send tokens and if you do not have gas or do not have enough, you cannot create transactions and therefore cannot send tokens. These gas shortage issues add to the complexity of use and have been a barrier to widespread blockchain adoption, and the risk of exposing and losing private keys due to changes or loss of the device that created the wallet has also been passed on to the user, limiting widespread adoption of blockchain-based distributed ledgers [4,5].

This paper proposes the use of a key management module of a server-based blockchain wallet to securely store private keys in keystore files and databases, effectively minimizing the risk of loss, and since the web UI is used even when changing devices, there is no requirement to move private keys, so there is no risk of private key exposure. In addition, by implementing the gas auto-charging function of the token transfer module, it automatically charges and transfers transactions when the gas is low, allowing users to transact smoothly without the burden of managing gas costs. The system's transaction verification module, Transaction Explorer, utilizes transaction ID to ensure transaction integrity and non-repudiation without exposing personal information [6]. It also introduces advanced security measures such as blocking overseas access IPs and DDoS defense to safeguard user data. It aims to provide a P2P online payment blockchain system that the public can use conveniently and safely by implementing an identity authentication function using smartphones and a platform-independent web application [7].

This paper is organized as follows: In Section 2, we review existing research and solutions in blockchain technology that can address the challenges and issues faced in blockchain transactions. Section 3 introduces our proposed face-to-face, peer-to-peer real-time token payment system. In Section 4, we demonstrate the implementation principle of the main functions of the system through implementation, and in Section 5, we verify the operation of the service through performance testing and analyze the implemented functions and performance. Finally, we evaluate the proposed system and conclude in Section 6.

## 2. Related Works

This chapter introduces ERC-20 and MetaMask as the underlying technologies of existing blockchains proposed to solve the problem of non-face-to-face transactions and examines their limitations. It also describes the underlying technologies used to develop the application proposed in this thesis and explains the improvements made to improve the limitations of the existing underlying technologies.

### 2.1. ERC-20

Ethereum's ERC-20 (Ethereum Request for Comment 20) was first proposed by Vitalik Buterin as an API standard for creating, managing and using tokens [8]. The purpose of the ERC-20 API was to create a standard interface for blockchain users to create tokens and send and receive them [9]. It is typically used in wallets such as MetaMask. The ERC-20 API consists of six mandatory functions (totalSupply, balanceOf, transfer, transferFrom, approve, allowance) and two events (transfer, approval), as listed in Table 1 [10].

**Table 1.** ERC-20 API: Methods.

| | Name | Description |
|---|---|---|
| Method | totalSupply | Provide the total amount of tokens |
| | balanceOf | Querying token balances |
| | transfer | Transfer Tokens |
| | transferFrom | Send tokens on a proxy |
| | approve | Delegate the right to withdraw your own tokens to spenders |
| | allowance | Allowance returns the number of tokens a spender can proxy withdraw from an owner |
| Event | transfer | Emitted when value tokens are moved from one account (from) to another (to). |
| | approval | Emitted when the allowance of a spender for an owner is set by a call to approve |

This approach uses smart contracts that run on the Ethereum EVM, written in the programming language Solidity, compiled and registered on the Ethereum blockchain via transactions [11]. Creating and managing multiple tokens requires a lot of work, as you need to develop and manage solidities individually and there are limitations that make it difficult to manage as there are no guarantees on behavior. To improve these issues, this paper presents an easy way to create ERC-20-like tokens using SCT-20 proposed by the Symverse blockchain [12].

*2.2. MetaMask*

In blockchain, a wallet is an application designed to generate, store and manage private keys and to send and receive coins and tokens, which are assets on the blockchain. Once a user creates a private key through their cryptocurrency wallet, they can generate a public key and the hash value of the public key creates an address on the blockchain. Transactions can be made through this address and all transactions are recorded on the blockchain network and can be viewed through the explorer. MetaMask is a web browser plugin application used by most users as a cryptocurrency wallet for storing ERC-20-based tokens that operate on top of Ethereum and the Ethereum blockchain [13]. MetaMask allows you to store and trade Ethereum, integrate MetaMask into decentralized financial applications and manage and trade various tokens and NFTs [14]. MetaMask stores wallet settings, transaction data and more in the form of LevelDB [15]. The path to the LevelDB in the most popular browsers Chrome and Edge is shown in Table 2. In Table 2, PROFILE refers to the user profile set in the browser, which has the value "Default" when the user does not use multiple profiles. The ID is a unique ID given to the browser extension, with different values depending on the browser: "nkbihfbeogaeaoehlefnkodbefgpgknn" for Chrome and "ejbalbakoplchlghecdalmeeeajnimhm" for Edge.

**Table 2.** Path of MetaMask LevelDB folder.

| Browser | Path |
|---|---|
| Chrome | %USERPROFILE%\AppData\Local\Google\Chrome\User Data\{PROFILE}\Local Extension Settings\{ID} |
| Edge | %USERPROFILE%\AppData\Local\Microsoft\Edge\User Data\{PROFILE}\Local Extension Settings\{ID} |

MetaMask exposes your private key when you change devices and while the "clear browsing data" feature does not delete the extension's data, removing the extension directly from the browser deletes the entire folder containing LevelDB, which is irreversible and irreparable [16]. To improve this problem, this paper does not store the private key on the device, but on the server's keystore for safe storage.

### 2.3. Limitations of Existing Solutions

A blockchain wallet is used to store private keys, generate signatures and create transactions on behalf of users. In short, it is an application that allows you to make real-time payments on the blockchain [17]. If we divide wallets into categories, the first case is web wallets, which are available through any web browser and are convenient and easy to use. The web interface has compatibility issues with mobile devices such as smartphones and is vulnerable to DDoS attacks over the Internet. In addition, it is necessary to secure the user's key due to the exposure of the private key when moving the key [18]. In the second case, when using web browser extensions such as MetaMask, the problem of deleting private keys that are difficult to recover due to browser deletion or extension deletion is the biggest blind spot as pointed out in Section 2.2 [16]. In the third case, a mobile wallet using a smartphone is highly portable but puts the user in control of and makes them responsible for their keys. The worst-case scenario for the security of your keys is that if you lose your smartphone, you will not be able to get them back, or if you change your smartphone, your keys will be exposed to the outside world and you will have a hard time figuring out how to move them [19]. In the fourth case, if you use a desktop wallet, your private keys are either stored on your PC or hosted by a third party, such as a service provider. The disadvantages of storing your keys on your PC are that they may be compromised or deleted and if they are hosted by a company that offers customization services or hardware wallet services, or you may have to pay for them [20,21].

The wallets listed above are storage wallets used to store coins or tokens. Therefore, it is inconvenient to manage the coins used as gas to transfer tokens and there is also a difficulty in purchasing and transferring through the exchange when there is a shortage of gas. This causes great inconvenience in the use of payment transactions between individuals and is a major obstacle to blockchain activation. In this paper, we implemented the core function of automatic gas charging in the token transfer module in Section 3.4, so that gas is automatically charged when transferring tokens, so that token transactions between individuals can be carried out without the need to manage gas, eliminating the obstacles to token transactions.

In addition, a web-app wallet is proposed so that various types of wallet applications can be applied to various devices (smartphones, tablets, PCs, etc.) [22] and this web-app wallet is written with the Flutter framework and can be used with the same interface on all devices and is excellent for obfuscation and security [23]. The proposed web-app wallet does not expose keys when changing wallets due to device changes and does not lose keys when the device is lost and defends against DDoS attacks [24] that are vulnerable to server-based wallet services by using the 3.5 DDoS protection module and protects against key loss by creating wallets using low-capacity keystores such as List 1 in the 3.2 Key management module and by storing secondary backup copies in the DB.

### 2.4. Background Technology

#### 2.4.1. Ethereum Keystore

An Ethereum keystore is a means of authenticating to a specific address on Ethereum and is a file that stores a private key encrypted with a passphrase [25,26]. To obtain a private key, you need to know not only the keystore file but also the passphrase, and, for usability reasons, it is a key storage standard created for secure transactions using the keystore and passphrase combination rather than exposing the private key and using it directly [27].

- Generating a Keystore

The Ethereum platform generates a private key and a public key using the ECDSA (Elliptic Curve Digital Signature Algorithm). The passphrase is encrypted using a one-way cryptographic algorithm called "Scrypt" to generate a derived key, as shown in Figure 1 [28].

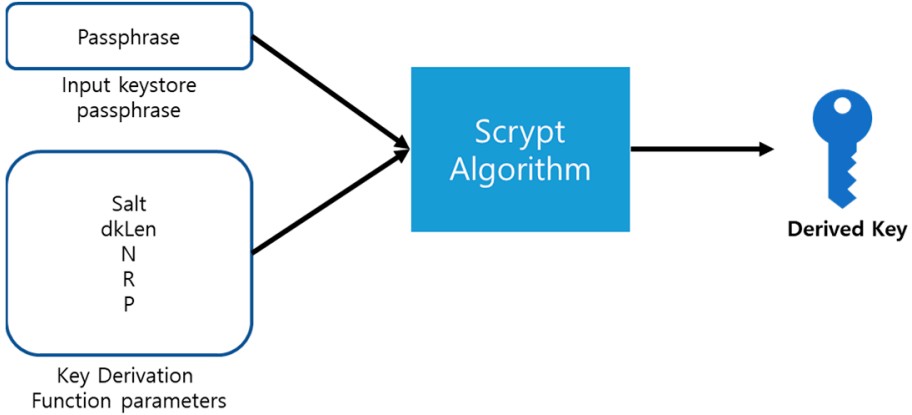

**Figure 1.** Generate a derived key.

For the decryption of the private key, it is encrypted using the AES algorithm as shown in Figure 2 and then a cipher text needs to be generated.

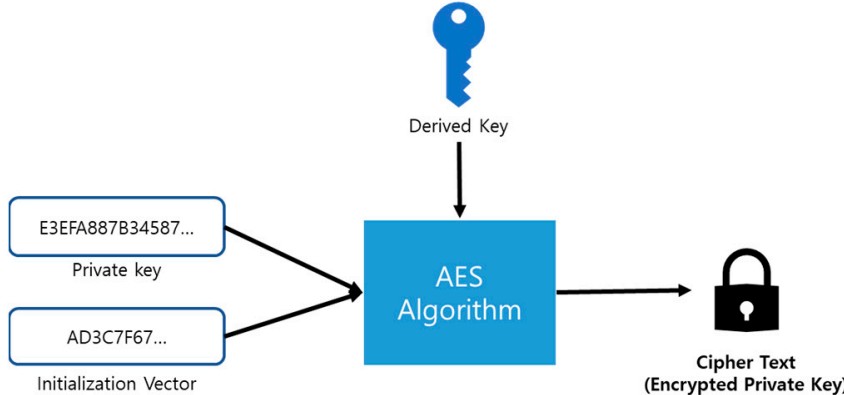

**Figure 2.** Create a cipher text.

The MAC for verifying whether the user-input passphrase matches is stored in the keystore by concatenating the last 16 bytes of the derived key (32 bytes) with the cipher text and hashing the result using the SHA3-256 hash function, as shown in Figure 3.

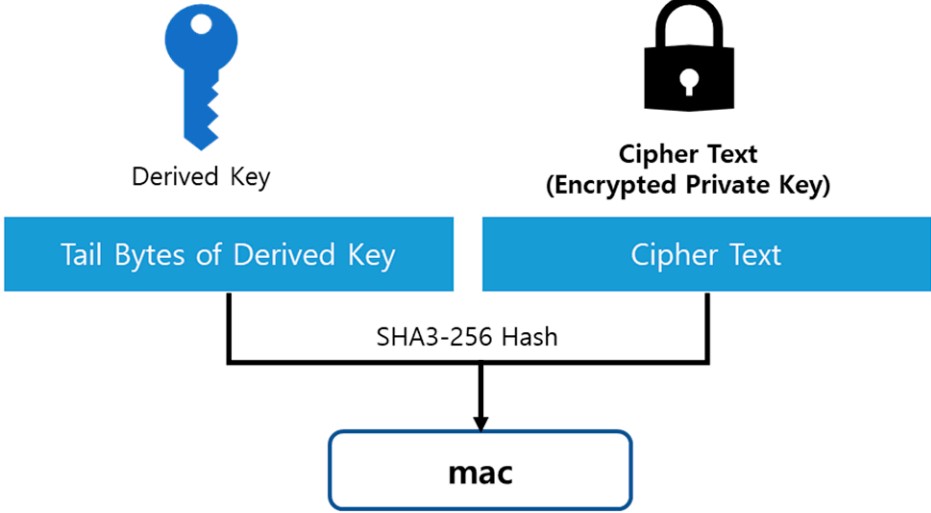

**Figure 3.** Create mac from derived key and cipher text.

The resulting keystore file created in this way is shown in Listing 1.

**Listing 1.** Keystore file example.

```
{
    "version":4,
    "id":"64cbaeb8-431b-41d4-a5e6-0508fc509b74",
    "address":"0x1bff0b319a73b51159fb4e2d0111d5c93fa1b3d6",
    "crypto":{
        "ciphertext":"1ab011aeae5d288465a1f4c89cf6b4a494ba90d908ef015ffa43fa9838ff1483",
        "cipherparams":{
            "iv":"2dd0641c64f19d978854a0ab3e27c0a8"},
            "cipher":"aes-128-ctr",
            "kdf":"scrypt",
            "kdfparams":{
                "dklen":32,
                "salt":"cfaeccc4d27f0305f0af2d3d87214a360a06714cdf2320db44c815b9c03b4ce8",
                "n":4096,
                "r":8,
                "p":1
                },
            "mac":"5ad80b19d7338245fb12129c2c441eee104ab054171edfd07c44cd602cdefdf6",
            "machash":"sha3256"
    }
}
```

- Decrypting Keystore

To decrypt the keystore, you must first verify that the entered passphrase is correct. Based on the entered passphrase, a newly derived key and MAC are generated and checked for a match with the MAC within the keystore. If a match is confirmed, the newly derived key, cipher text and cipher parameters information within the keystore are input into the AES decryption algorithm to decrypt the cipher text into the private key, as shown in Figure 4 [29].

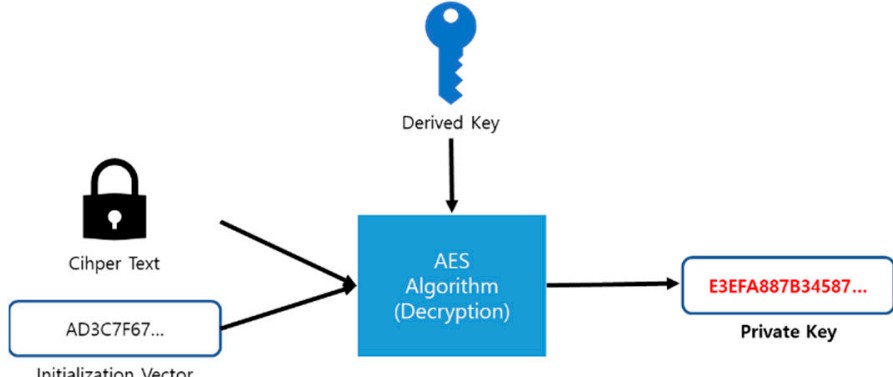

**Figure 4.** Decrypt cipher.

2.4.2. Symverse

Symverse is a layer-1 blockchain platform with one-second block finality based on self-sovereign distributed identities, using the SYM coin. It uses a unique ID system of 10 bytes consisting of a network identifier called SymID (2 bytes), a CitizenID (6 bytes) and an account identifier (2 bytes) to replace the approximately 20 bytes of public key hash used as an address. It is a scalable multi-blockchain platform as a cooperative blockchain service that interoperates between independent blockchain platforms based on the Symverse platform. In particular, the block generation method is an enhanced BFT (Byzantine Fault Tolerant) method with strategic voting, and fast block finality is implemented using PoS (proof of stake) [30].

- SCT-20

SCT (Symverse contract template) is a template protocol designed to make it easy to create and operate smart contracts within the Symverse blockchain. In the case of Ethereum, the ERC-20 protocol allows you to write a smart contract with Solidity and register it on

the blockchain through a transaction and operate the smart contract through the EVM [29], but in Symverse, SCT-20 is a template that provides standard inputs and outputs as shown in Table 3 to easily create a token smart contract on the blockchain with data type JSON via RPC [31]. To create an SCT contract, use the SCT_CREATE function in Table 4 and pay the transaction fee of 0.8049 SYM coins. The SCT contract generates a transaction using the RLP (recursive length prefix) encoded value of array ("0x14", 0, array ("SYMBOL NAME", "SYMBOL", convertEth2WeiHex (10,000,000,000, 18), SymID)) in the input parameter of Table 5, converted to Hex, and sends it to the Symverse blockchain to create a token smart contract. To transfer the generated token, we use the SCT_TRANSFER function in Table 3 and set the gasPrice in Table 4 to 7000 to transfer the token.

- Transactions

**Table 3.** SCT20 creation parameters.

| Parameters | Type | Description |
|---|---|---|
| Name | Address | Smart contract (token) name |
| Symbol | String | Smart contract (token) symbol. The length should be from 3 to 10 |
| Amount | Int | Total supply |
| Owner | Address | 10 bytes—address of the contract owner |

**Table 4.** SCT20 operation gas.

| Type | Function | Description | SCT Gas | Total Gas |
|---|---|---|---|---|
| SCT20 | SCT20_CREATE | Create SCT20 contract | 49,000 | 8,049,000 |
| | SCT20_TRANSFER | Transfer token | 7000 | 56,000 |
| | SCT20_TRANSFER_FROM | Token delegation transfer | 9000 | 58,000 |
| | SCT20_MINT | Issue additional token | 7000 | 56,000 |
| | SCT20_BURN | Token burn | 7000 | 56,000 |
| | SCT20_PAUSE | Contract suspension | 4000 | 53,000 |
| | SCT20_UNPAUSE | Resume suspended contract | 4000 | 53,000 |

**Table 5.** Transaction data parameters.

| Field | Type | Description |
|---|---|---|
| from | address | [10] bytes, sender address |
| nonce | int | the count of transaction publication in the account |
| gasPrice | int | gas price per gas unit |
| gas | int | gas amount for executing transaction |
| to | address | [10] bytes, receiver address or contract address or nil |
| value | int | the value sent with this transaction or amount of deposit |
| input | data | [] byte, rlp encoded data (contract or sct) |
| type | int | Transaction type (0: general (default), 1: sct, 2: deposit) |
| workNodes | [] address | array, list of work nodes that deliver the transaction (count = 1) |
| extraData | int | [] byte |

A transaction is an act of recording a ledger in a block on the blockchain, and once a transaction is recorded, it cannot be modified or deleted. In Symverse, transactions include normal transaction behavior, SCT transactions and deposit transactions [31]. The data required for a transaction is shown in Table 5.

The transaction of general transaction behavior is to transfer SYM coins, SCT transaction is to transfer smart contracts, and deposit transaction is for interest distribution. The gas consumed by a transaction is calculated using the following formula.

The gas consumed in a transaction is calculated using the following Formula (1):

$$
\begin{aligned}
\text{Consumed\_gas} = &\ \text{base\_gas} \\
&+ (\text{number of none-zero-byte}) * 680 \\
&+ (\text{number of zero-byte}) * 40 \\
&+ \text{contract\_operation\_gas}
\end{aligned}
\tag{1}
$$

Consumed gas is calculated as the base gas rate (base_gas) set in the blockchain plus the number of non-zero bytes of input_data*680 and the number of zero bytes*40 plus the contract operation gas rate.

The token transfer transaction (1) prepares a private key corresponding to the sender's SymID, (2) retrieves the recent transaction nonce in the SymID's blockchain and (3) prepares the raw transaction data as follows.

a. Add 1 to the recent transaction nonce. b. Prepare $input = array ("0x14", "0x1", array("Receiver SymID", convertEth2WeiHex(1000000000, 18))) to be used as transaction input and prepare. c. RLP-encode the sct_data d. Prepare the transaction data. $tx_req = array ("from" => "Sender SymID","to" => "Contract ID","gasLimit" => bcdechex(2000000), "gasPrice" => bcdechex(100000000000), "value" => 0x0, "nonce" => LatedNonce+1, "type" => "1", "workNodes" => array("Work Node SymID"), "input" => "0x".RLP(sct_data)) e. Sign the transaction data. (4) Send an RPC from the signed data as a JSON data type. $param_arr = array ("jsonrpc" => "2.0", "method" => "sym_sendRawTransaction", "params" => array($tx_raw), "id" => 1) Use the returned transaction hash to obtain a transaction receipt to check if the transaction was processed successfully. You can see the transaction processing flow in Figure 5.

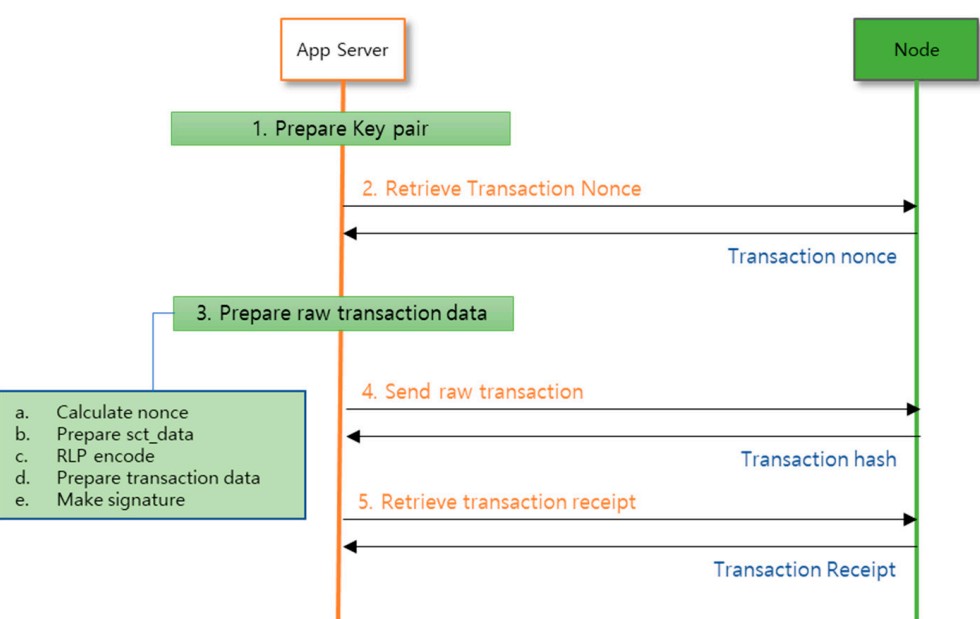

**Figure 5.** Symverse transaction processing flow [31].

The next chapter will introduce the details and architecture of the system proposed in the thesis and the various modules that address the problems discussed in this chapter. We will also provide an evaluation of the system's performance, security and usability, demonstrating its potential to provide a comprehensive solution for non-face-to-face financial transactions.

### 3. Token Payment Blockchain System

The proposed token payment blockchain system is a secure system that enables real-time token transfers between individuals or parties without exposing personal information. Once the transaction is completed, the transaction history can be verified by anyone using a wallet or using a separate blockchain scanner. This architecture consists of five modules: a module for wallet owners to generate and store private keys in the keystore, a module for checking and charging insufficient gas in the wallet, a module for sending and receiving tokens, a module for DDoS defense of the system and a module for real-name authentication of the keystore.

### 3.1. Overall Architecture

The blockchain system consists of a key management module, token management module, token payment module, DDoS defense module and real-name authentication module for the legacy system. The key management module efficiently generates, changes and retrieves private keys in conjunction with the keystore, while the token management module manages tokens by creating, querying, minting and burning them. The token payment module automatically charges and sends gas when a general user has no gas. The DDoS defense module is composed of distributed attack prevention and blocking of overseas access to protect the server and the real-name authentication module authenticates the wallet according to domestic law. Figure 6 shows the module-by-module configuration and interconnection of the system that enables P2P real-time payments based on the server.

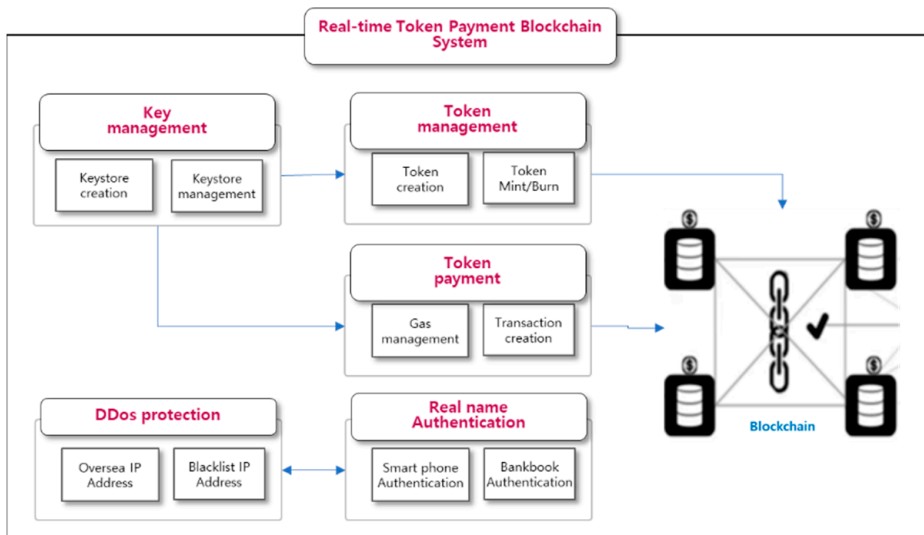

**Figure 6.** Token payment blockchain platform.

The control flow of the proposed system is shown in Figure 7. When the user's smartphone enters the blockchain address of the destination to be transmitted and the quantity of tokens to request transmission, the key management module reads and decrypts the user's keystore for transaction signing in the key management module, extracts the private key, signs the private key and sends it to the token transmission module. When paying for tokens, it checks the gas of SymID, the blockchain address of the sender and automatically charges the gas when it runs out and sends the transaction to the blockchain node. The transactions sent from the node to the blockchain are put into a block and when the block is confirmed and finalized through PoS (proof of stake) consensus, it is permanently distributed and stored on the blockchain. When the recipient's wallet requests a token balance inquiry, the balance is received by querying the node through the corresponding token management module.

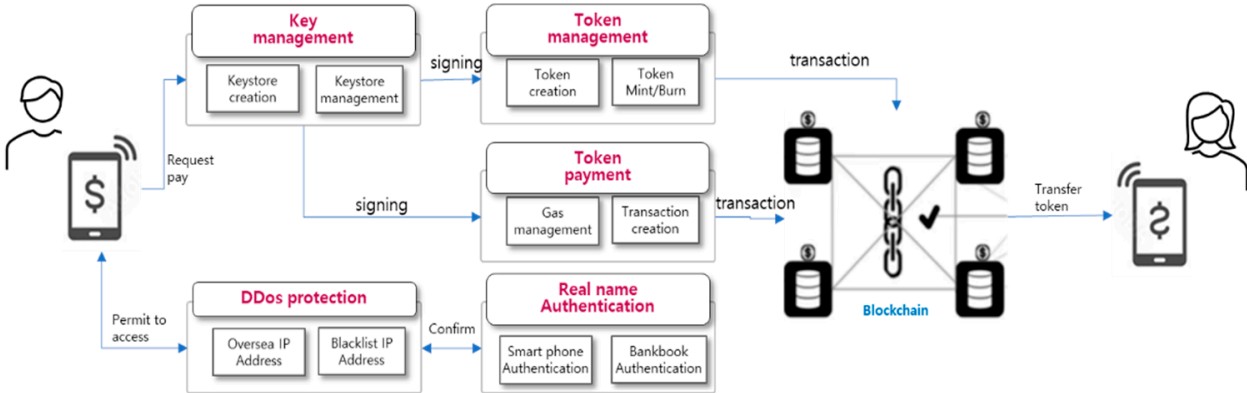

**Figure 7.** Control flow of system.

### 3.2. Key Management Module

Transaction creation process is illustrated in Figure 8. The key management module generates a private key as a 32-byte hexa code by creating a 256-bit random number when the user requests to create a wallet by entering a passphrase. It also generates a corresponding public key and public key hash and stores them in the file system as a keystore. The generated public key hash and the user's ID are sent to the certificate authority server of the Symverse blockchain to create a SymID, which is an address used in the Symverse blockchain, and store it in the DB. Users can send tokens with just their ID and passphrase at login without having to remember their SymID. The passphrase for the keystore can be changed, but the private key and SymID cannot be changed. When a user requests to sign a transaction raw code, they enter the passphrase to decrypt and extract the private key from the keystore, sign the raw code and create a transaction.

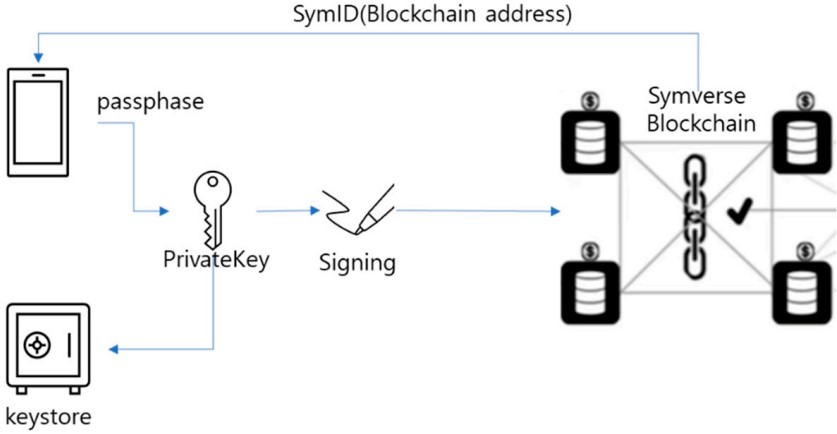

**Figure 8.** Transaction creation process through private key signature.

### 3.3. Token Management Module

Token management module is illustrated in Figure 9. The token management module is responsible for creating tokens that represent the initial supply quantity and can mint or burn the token amount as needed. Token minting on the Symverse blockchain can be performed using the SCT-20 template, which has a minting function to increase the total token issuance amount and a burn function to decrease it. To create a token, you need to input the token symbol name, symbol, total issuance quantity, owner SymID according to the JSON convention of SCT-20 and sign the generated bytecode with the private key extracted from the keystore using a passphrase. The bytecode is then converted into a transaction and permanently registered on the blockchain. To retrieve information such as the token contract address, you can query the blockchain using the hash value of the returned transaction receipt.

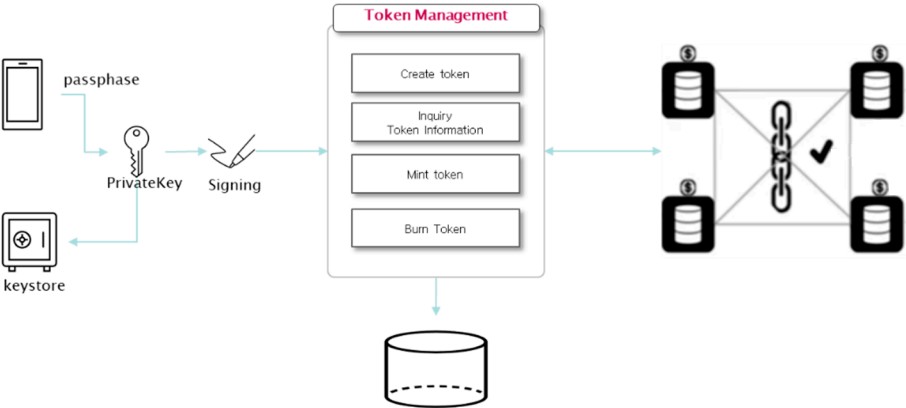

**Figure 9.** Token management module.

### 3.4. Token Transfer Module

Token transfer module is illustrated in Figure 10. The token transfer module is responsible for transferring tokens between users. When a user sends tokens, gas is consumed to execute the smart contract and the user must manage the gas separately, making it difficult to use. To make it easier for users, a module is provided so that it manages the gas consumption payment instead of users. To transfer tokens, the user generates bytecode and signs it using their private key. The gas consumed by the transaction is then calculated. If there is not enough gas, the contract owner requests that SYM coins have to be automatically recharged to the SymID account on the blockchain. In the next step, the transaction is sent to the node and permanently recorded in the block through a consensus algorithm so that the tokens are transferred to the recipient.

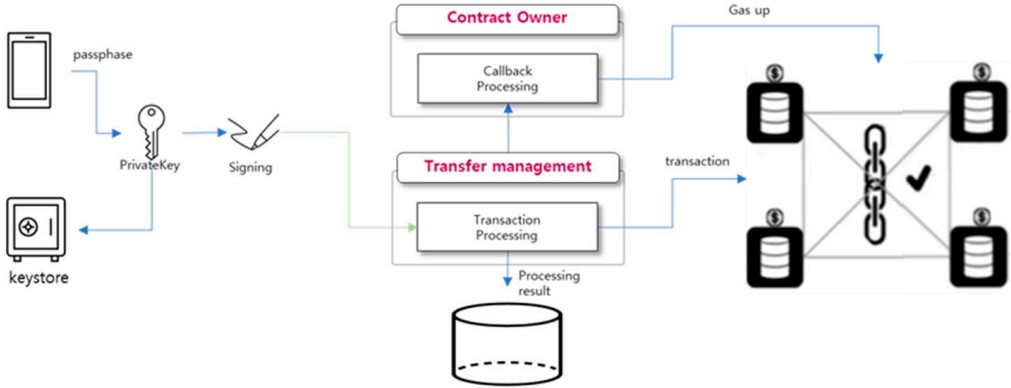

**Figure 10.** Token transfer module.

### 3.5. DDoS Protection Module

DDoS block module architecture is illustrated in Figure 11. The current token payment system has been developed for the purpose of actual use in Korea, so the module aims to block overseas IPs so as to comply with Korean laws for real-name authentication and to indiscriminate attacks from overseas. Based on the built-in database of global IP address allocation, it extracts IP addresses allocated to Korea and allows access only if they are included within the Korean IP address range. It also manages DDoS source IPs separately in the database to block access and effectively defends against DDoS by handling traffic from the built-in database without traffic increase on the main DB during multiple loads.

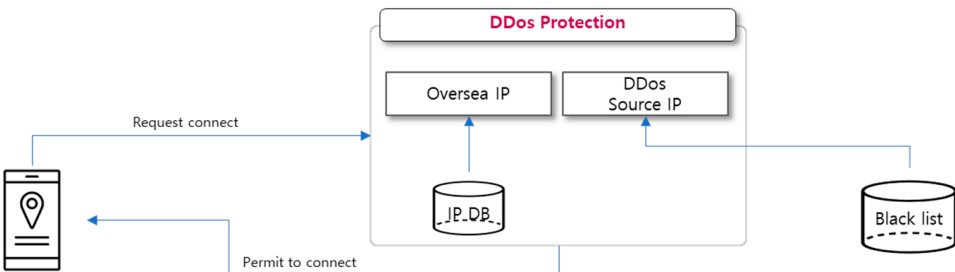

**Figure 11.** DDoS block module.

### 3.6. Real Name Verification Module

Real-name authentication module architecture is illustrated in Figure 12. The real name verification module interfaces with external legacy systems. When a smartphone user enters their name, phone number, date of birth (six digits) and the first digit to denote the sex code in the resident registration number, the server converts it into the format requested by the related institution and sends a response code to the smartphone number via SMS (short messaging system). The user inputs the response code and the module verifies their real name by receiving a response from the credit information management institution and performs real-name verification for the keystore.

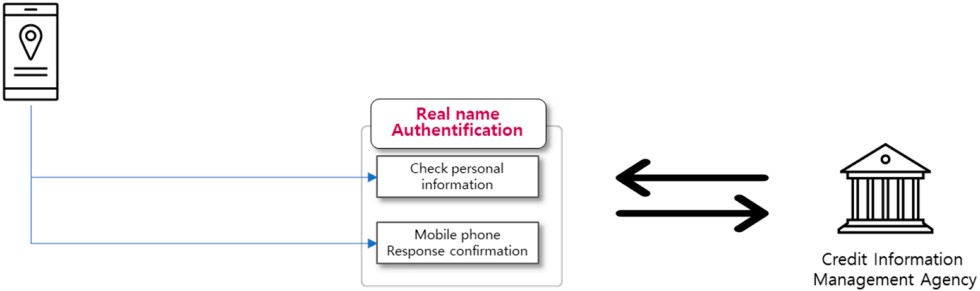

**Figure 12.** Real-name authentication module.

## 4. Implementation

The proposed system is composed of the commonly used server modules, CentOS and Apache, PHP and mySQL. The overseas IP blocking database is separately configured using SQLite. The blockchain is integrated with the Symverse mainnet through a service node installed for users' convenience. In particular, the user interface is programmed by using the Flutter language framework provided by Google so that it enables various devices such as smartphones, tablets and PCs to utilize the proposed system.

### 4.1. System Environment

The implementation system consists of a web service server and a blockchain work node called gsym. The web service is installed with CentOS as the operating system, Maria DB as the database, Apache as the web server and PHP as the scripting language to implement the REST server. The user interface is developed in Flutter language using Android Studio (Version 11.0.12+7) and is released as a web version uploaded to the server for users' convenience through web browsers. The database for blocking overseas IPs is implemented using Sqlite3 (Version 3.31.1) to separate it from the main database so that it does not affect the internal logic. The list of environments and libraries used in the implementation is shown in Table 6.

**Table 6.** List of environments and libraries.

| Environments | Specifications | Environments | Specifications |
|---|---|---|---|
| OS | Centos 7.4.17 | Database | mySQL 5.7.36 sqlite3 |
| Language | PHP/HTML/Dart/Javascript | Webserver | Apache 2.4.52 |
| Webserver-plugin | Php 7.4.1 Scrypt 1.4.2 | Blockchain | symverse |
| UI framework | Flutter/Bootstrap | | gsym |
| Test tool | JMeter V5.5 | Browser | Chrome version 111.0.5563.66 (build) (64bits) |

### 4.2. Wallet Creation

To use the wallet, generate a private key for each user who has completed smartphone authentication, save it as a keystore in the filesystem and send the key hash to the Symverse CA server (https://mainnet-ca.symverse.com/ca/v1/citizenInfo, accessed on 15 May 2023) to generate a SymID by sending JSON data {"id": {number},"mobileNum": "{smartphone number}", "publicKeyHash": "{key hash}", "userNm": "{user name}", "verificationType": "CELL_PHONE"} as a POST request and obtain back the generated SymID and store it in the user table to map the user ID and SymID. In this case, the keystore stored in the file system is restored in the backup table corresponding to the SymID, so that the keystore in the file system is redundant. This is used as a backup in case of accidents such as bulk deletion of keystores stored on the server.

### 4.3. Token Creation

The smart contract used in this paper is created by the smart contract by creating a transaction using the Listing 2 Array of SCT-20 input parameters and sending it to the Symverse blockchain. In other words, instead of writing smart contract code in Solidity such as Ethereum, it creates an SCT-20 contract on the Symverse blockchain using the SCT-20 contracts and transactions described in Section 2.4.2. As a result, a smart contract ID of "0x4801e91a5068757a9484" was created, link (https://scan.symverse.org/v2/sct2x_token/SCT20/0x4801e91a5068757a9484, accessed on 15 May 2023).

**Listing 2.** Array of SCT-20 input parameters.

```
sct_data = [
  "0x14", // SCT20
"0x0", // SCT20_CREATE Command
[
    "eSportsToken", // Token Name
    "EST", // Token Symbol
    "0x204fce5e3e25026110000000",   // Token Amount   (1 Billion)
    "0x00035b875ec2c5410002" // Owner's SymID
  ]
]
```

The smart contract ID is a required parameter when searching for the corresponding token or inputting a transaction according to the transaction; Sct_data is encoded with RLP (Recursive Length Prefix) as shown in Listing 3.

**Listing 3.** Sct_data RLP encoding result.

```
0xec1480e98c6553706f727473546f6b656e834553548c204fce5e3e250261100000008a00035b875ec2c54100
```

If you put this as the input value of Table 5 and sign the array of Table 5 and send it to the work node as a transaction, the block is confirmed and you can see that the token is created as shown in Listing 4.

**Listing 4.** SCT-20 information.

```
(
    [jsonrpc] => 2.0
    [id] => 1
    [result] => Array
        (
            [creator] => 0x00035b875ec2c5410002
            [name] => eSportsToken
            [owner] => 0x00035b875ec2c5410002
            [state] => active
            [stateCode] => 0x0
            [symbol] => EST
            [total] => 0x204fce5e3e25026110000000
            [type] => sct20
        )
)
```

In addition, another token can be created in the same way and the implementation system is designed to store the smart contract ID in the DB as a separator so that all token transactions can be operated.

*4.4. Balance Inquiry*

This is the most used feature of the wallet and it calls the balance corresponding to the blockchain address, SymID. The data flow for the balance inquiry is shown in Figure 13, and the client to make the call is a web-app developed using Flutter: (1) the web-app completes authentication for the user ID; (2) the web-app returns the balance for the user ID as JSON-formatted data {"act":"actGetBalance", "domain":"NooWeb", "user_id":"doltwo"} as an asynchronous HTTP request; (3) a REST API written in PHP that first checks for DDoS protection on the server and then looks up the SymID DB corresponding to the authorized user ID; (4) request this as JSON data {"jsonrpc":"2.0", "method":"src_getContractAccount", "params":["0x0ced1024eed02b234df2","0x00000000000000000001","lastest"], "id":1} to the Symverse node gsym using RPC; (5) the node responds with {"jsonrpc":"2.0","id":1,"result":{"balance": 10000000000000}}; and (6) sends this returned data to the webapp as JSON as {"est_id":"0x00000000 000000000001","balance": 10000000000000}} to display the SymID address and balance in the web-app.

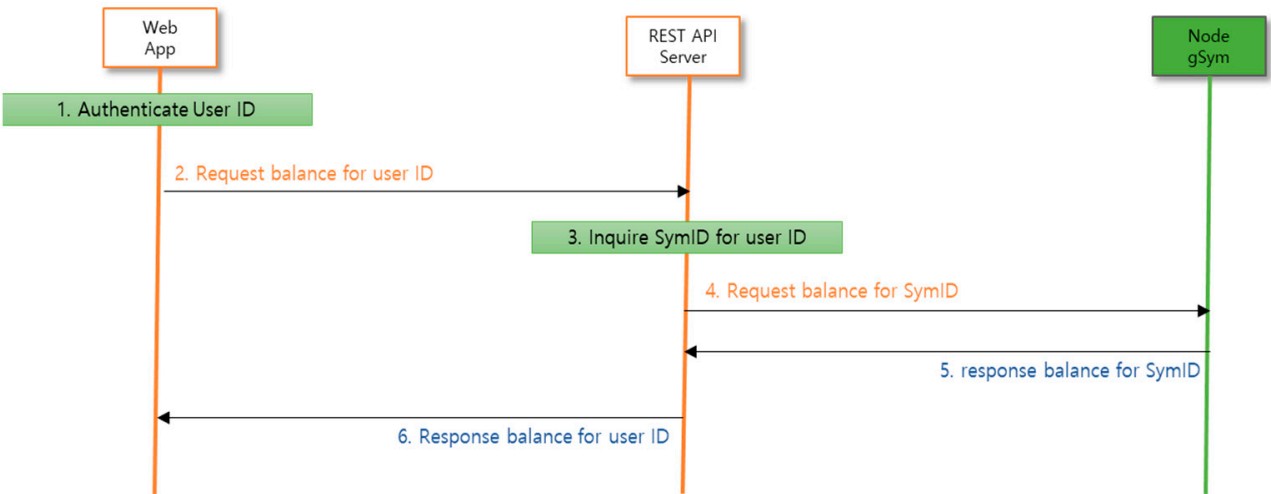

**Figure 13.** Control flow diagram for a balance inquiry.

*4.5. Token Transfer*

Token transfer is a user transfer function and is shown in Figure 14 as a diagram. (1) A sender with a user ID token authenticated through a web-app written in Flutter checks the recipient's SymID by looking up the phone number or name and adds the number of tokens and (2) requests a token transfer to the REST API server with JSON type data. (3) The server looks up the sender's SymID, (4) checks the balance of the SymID, (5) checks if the sender has more than the number of tokens to transfer and (6) creates a raw transaction through the private key management module, signs it and requests transfer to the token transfer module. (7) Check the gas of the SymID of the sender of the transmitted raw transaction and if there is insufficient gas to transmit, (8) send an amount calculated by a formula from the token transfer module to the blockchain node to automatically charge the sender's gas, (9) confirm the receipt for this operation and (10) send the

raw transaction data to the blockchain node, gysm. (11) Retrieve the receipt for the transaction and (12) send the result to the web-app.

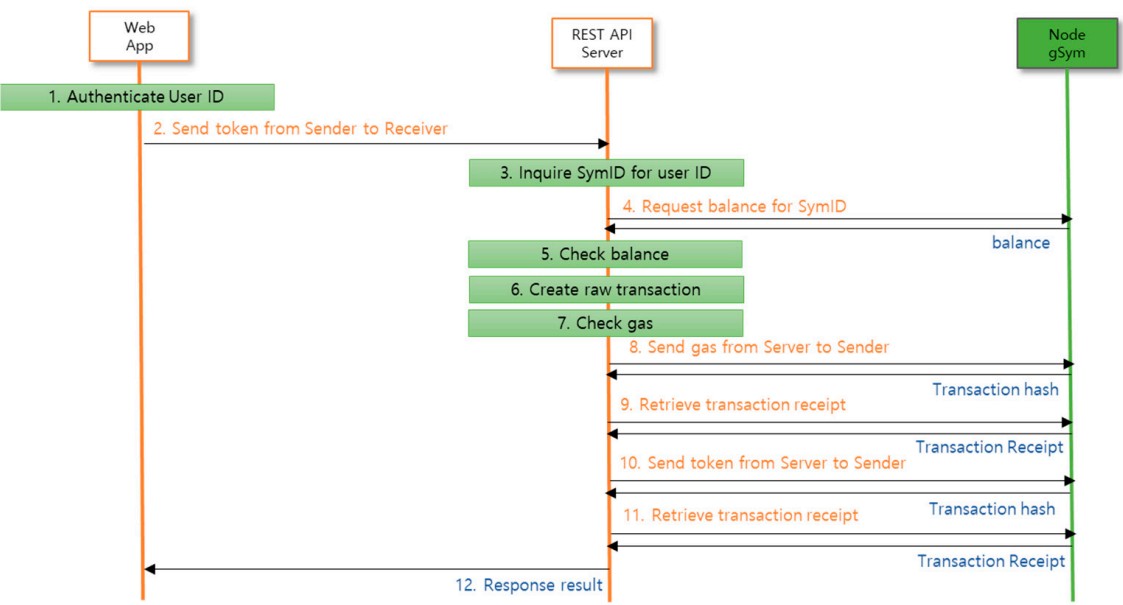

**Figure 14.** Control flow diagram for send token.

The token transfer module automatically recharges SymIDs that are low on gas, eliminating transfer errors due to low gas and allowing users to transfer tokens without considering gas.

### 4.6. DDoS Protection

DDoS defense consists of two components: a part that checks the nationality of the IP and blocks access if it is not an authorized nationality and a DB that maintains a blacklist of IPs for brute force attacks on the web server and blocks access. In this case, the nationality and blacklist for IP use sqlite3, which is a separate DB separated from the main DB, and only sqlite3 works in case of an accidental attack to protect the main DB, mySQL. Specifically, when a web app makes a connection, the REST API server can determine the connection IP using the PHP environment variable $_SERVER['REMOTE_ADDR']. For the corresponding nationality search, use the SQL query in Listing 5 to check the results of the table of IP bands for each country entered in sqlite3.

**Listing 5.** Geolocation query with IP lookup.

```
SELECT * FROM global_ip WHERE startipL <= ". ip2long({$_SERVER['REMOTE_ADDR']}). " AND endipL >=
". ip2long({$_SERVER['REMOTE_ADDR']})
```

Block international access using the country code and allow only local users. In addition, check the access IP $_SERVER['REMOTE_ADDR'] against the blacklist and allow only if it is not applicable. DDoS protection imposes a load on the server by investigating every connection and we will cover load testing for wallet creation and balance retrieval in Chapter 5 of this paper.

### 4.7. Database

The database consists of tables for user information, wallet and transfer, as well as tables for tokens, real-name authentication logs and IPs for DDoS defense. Table 7 summarizes the database table list for system implementation to operate a service.

**Table 7.** List of database tables.

| Table Name | Table ID |
|---|---|
| User | nr_user |
| Wallet | est_wallet |
| Transfer | nr_charge |
| Charge | nr_deposit |
| Withdraw | nr_withdraw |
| Statics transfer | nr_settle_sale |
| Statics join | nr_stat_join |
| Realname log | nr_log_sms |
| DDoS ban ip | nr_ddos_ip |

*4.8. User Interface of Client Side*

The user interface of the client used by wallet users is organized as shown in Table 8.

**Table 8.** User's menu.

| Menu | | Description |
|---|---|---|
| **Class1** | **Class2** | |
| Client | Login | Login with member ID and password |
| | HP authentication | Mobile phone real name verification |
| | Charge | Token charging function for payment |
| | Charge list | Recently charged history list |
| | Transfer | Transfer listings for sending to sellers |
| | Transfer list | List of sent history records |

The interface for the wallet user who receives payment is composed of menus that allow them to view transaction history (as in Table 9) and exchange the received tokens into fiat currency.

**Table 9.** Seller's menu.

| Menu | | Description |
|---|---|---|
| **Class1** | **Class2** | |
| Seller | Transfer list | Deposits and withdrawals of tokens and error history |
| | Convert list | Details of application for conversion and details of results |
| | Convert application | Application for changing safely traded tokens to fiat |

The platform operator's interface is configured with menus such as statistics management, member management and deposit/withdrawal management, as shown in Table 10.

**Table 10.** Platform operator's menu.

| Menu | | Description |
|---|---|---|
| **Class1** | **Class2** | |
| Platform | Statistics | Member registration status by time/day/month |
| | Member management | Member list/real name authentication management |
| | Settlement management | Settlement information/daily settlement/settlement statistics |
| | Deposit and withdrawal management | Conversion list/charging list/real-time notification |
| | Mobile gift certificate management | Mobile gift certificate sales list/purchase list |
| | Manager management | Admin list and administrator privilege setting |

To convert fiat currency to tokens in order to transfer tokens, users make a request for conversion through an interface as depicted in Figure 15a, which shows the actual usage example.

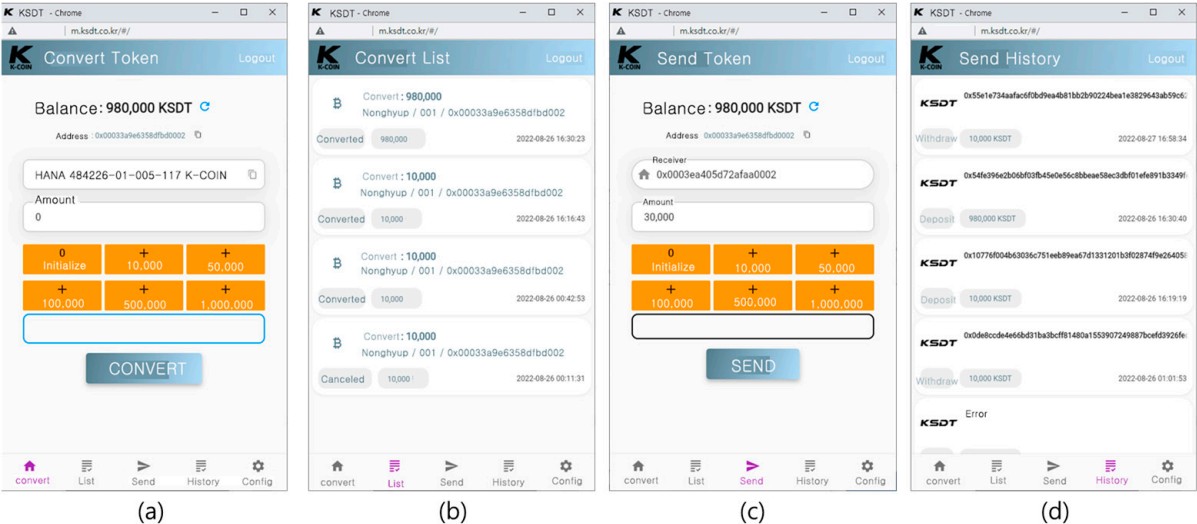

**Figure 15.** UI of P2P payment system (**a**) interface of convert token. (**b**) History of convert status. (**c**) Interface of transfer token. (**d**) History of transfer token.

The history of the conversion transaction can be confirmed in the deposit history as shown in Figure 15b and it can be checked whether the conversion has been completed.

After depositing tokens, users can transfer tokens to a specified seller after confirming the transfer address, using the Figure 15c interfaces.

All the transaction history (deposit/convert/payment) is displayed as a list as shown in Figure 15d to represent the result of the transaction.

### 4.9. External Interface

External interfaces are represented by three interfaces as shown in Table 11 and each interface communicates using the JSON RPC (Version 2.0) method.

**Table 11.** External interfaces.

| I/FID | Sender | Receiver | Interface Description | Data Format |
|-------|--------|----------|-----------------------|-------------|
| IF_001 | KSDT server | gsym | Symverse work node | JSON |
| IF_002 | Credit rating agency | Ok-names | Ok-names real name inquiry | JSON |
| IF_003 | Symverse | Symverse CA | Issue SymID | JSON |

## 5. Experiment and Results

To evaluate the proposed system, we measured it using JMeter (Version 5.5), which uses the BMT tool to test the HTTP protocol.

### 5.1. System Environment

Experiment environment architecture is illustrated in Figure 16 and Test environment is shown in Table 12. For performance evaluation, the system environment consists of a physical node with a main web server environment with one CPU (AMD Ryzen 5 PRO 3400GE) and 8GB DDR4 main memory and an SSD of 240GB, a physical node as a Symverse work node with server environment consisting of one CPU (Intel(R) Xeon(R) CPU E3-1230 v6 @ 3.50GHz) and 8GB DDR4 main memory and an SSD of 240GB and a client system with one CPU (Intel(R) Core(TM) i7-8750H CPU @ 2.20GHz) and DDR4 16GB memory so that it can perform the evaluation of various experiments.

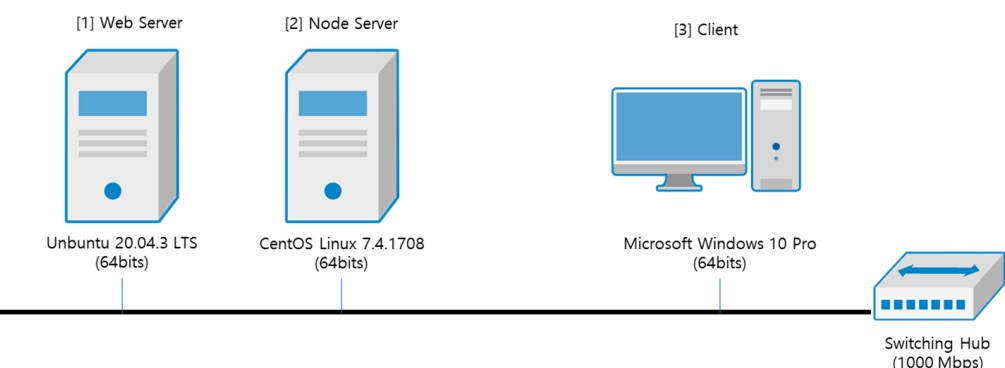

**Figure 16.** Experiment environment.

**Table 12.** Test environment.

| No | Role | OS | CPU | Memory | Disk | Spec |
|----|------|-----|------|--------|------|------|
| 1 | Web Server | Ubuntu 20.04.3 LTS | AMD Ryzen 5 PRO 3400GE | 8G | SSD 256GB | -mySQL 5.7.36 |
| | | | | | | -Apache 2.4.52 |
| | | | | | | -PHP 7.4.1 |
| | | | | | | -sqlite3 |
| 2 | Node Server | CentOS Linux release 7.4.1708 | Intel(R) Xeon(R) CPU E3-1230 v6 @ 3.50GHz | 8G | SSD 512GB | -gSym |
| 3 | Client | Windows 10 Pro(64bit) | Intel core i7-8750H 2.20Ghz | 16G | SSD 512GB | -chrome V111.0.5563.65 |
| | | | | | | -jMeter v5.5 |

*5.2. Performance Testing*

The token payment system is a system that provides services through JSON format using RPC, not web page calls, as it is composed of a web application server. We implement performance testing on wallet creation response and balance inquiry, where both systems call and the results are representing a simple JSON format.

5.2.1. Wallet Creation Response Time

Wallet creation requests are sent to the server and the results are stored in the database. First, the results up to keystore creation are tested. Then, to measure the pure wallet creation time, the IP blocking for DDoS prevention is turned off. For the user10 scenario, assuming a loop of 100 requests with 10 threads, the ramp-up is set to 0, allowing only up to 10 requests per second. For the user20 scenario, assuming a loop of 100 requests with 20 threads, the ramp-up is set to 0, allowing only up to 20 requests per second. For the user50 scenario, assuming a loop of 100 requests with 50 threads, the ramp-up is set to 0, allowing only up to 50 requests per second. The summary of the test results for the three scenarios is shown in Table 13.

**Table 13.** TC1 Response time of keystore creation (without DDoS protection).

| Requests | Requests | | | | | | Response Times (ms) | | | | Throughput | Network (KB/s) | |
|----------|----------|----|--------|-------------|-----|------|--------|------------|------------|------------|----------------------------|---------|------|
| Label | # of Samples | KO | Error% | Average | Min | Max | Median | 90th pct | 95th pct | 99th pct | Transaction (Per sec) | Receive | Send |
| **Total** | **8000** | **2** | **0.03%** | **1222.38** | **11** | **2037** | **1568.00** | **1732.00** | **1767.00** | **1829.00** | **27.98** | **10.37** | **6.07** |
| User10 | 1000 | 0 | 0.00% | 322.70 | 230 | 552 | 314.50 | 404.00 | 404.00 | 450.96 | 30.66 | 11.35 | 6.59 |
| User20 | 2000 | 0 | 0.00% | 646.74 | 210 | 912 | 650.50 | 800.95 | 800.95 | 861.00 | 30.52 | 11.30 | 6.56 |
| User50 | 5000 | 2 | 0.04% | 1632.57 | 11 | 2037 | 1646.00 | 1784.00 | 1784.00 | 1843.00 | 30.47 | 11.30 | 6.54 |

The average response time corresponding to the variation in the number of threads is depicted in Figure 17a.

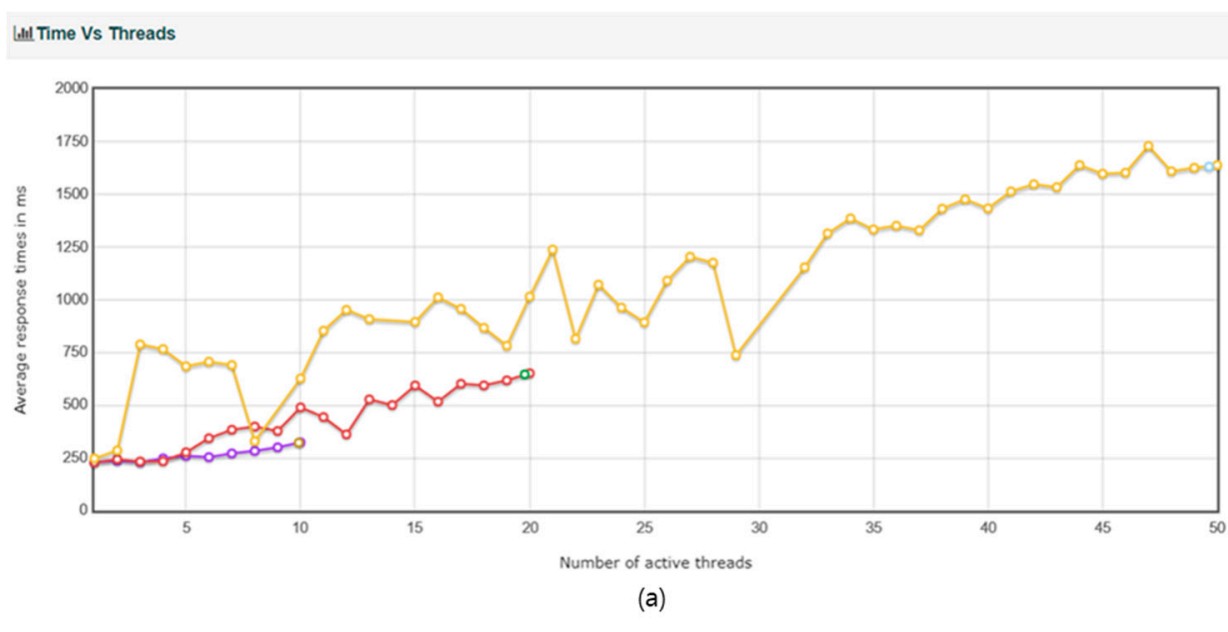

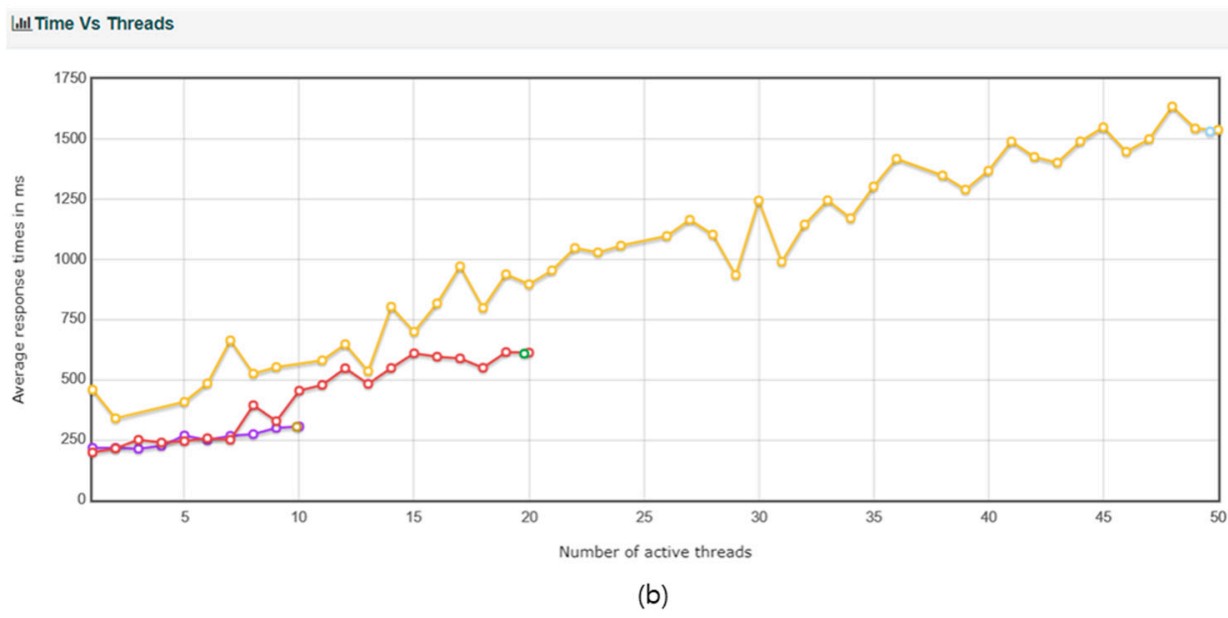

**Figure 17.** TC1 active number of threads vs. response time (**a**) without DDoS protection (**b**) with DDoS protection. (Puple: User10, Red: User20, Yellow: User50).

Under the same conditions, the results of processing with DDoS blocking turned off are shown in Table 14 and the graphical representation of the results is shown in Figure 17b. The DDoS blocking logic showed an impact of approximately two transactions per second, with 1.61 for user10, 1.9 for User20 and 2.0 for user50 in terms of transactions per second. However, for user50, there was a significant variation in response time.

**Table 14.** TC2 Response time keystore creation (with DDoS protection).

| Requests | Requests | | | Response Times (ms) | | | | | | | Throughput | Network (KB/s) | |
|---|---|---|---|---|---|---|---|---|---|---|---|---|---|
| Label | # of Samples | KO | Error% | Average | Min | Max | Median | 90th pct | 95th pct | 99th pct | Transaction (Per sec) | Receive | Send |
| **Total** | **8000** | **2** | **0.10%** | **1148.02** | **6** | **1891** | **1483.00** | **1612.00** | **1641.00** | **1700.00** | **17.00** | **6.32** | **3.65** |
| User10 | 1000 | 0 | 0.00% | 304.58 | 209 | 488 | 296.00 | 360.90 | 390.00 | 431.98 | 32.27 | 11.94 | 6.93 |
| User20 | 2000 | 0 | 0.00% | 608.69 | 191 | 825 | 617.00 | 711.00 | 737.00 | 786.99 | 32.42 | 12.00 | 6.97 |
| User50 | 5000 | 8 | 0.16% | 1532.44 | 6 | 1891 | 1544.00 | 1633.00 | 1661.00 | 1712.99 | 32.47 | 12.11 | 6.97 |

In Figure 17a,b, the difference in results is due to the same condition, but when DDoS blocking is turned off and when DDoS blocking is not turned off, the result of transactions per second is different, such as 1.61 for user10, 1.9 for user20 and 2.0 for user50, as shown above. However, the difference between the two conditions is not significant, which shows that the performance of the proposed technique is excellent.

### 5.2.2. Balance Inquiry Response Time

Balance response is the process of querying SymID from the blockchain node and delivering the balance to the client in JSON format. To measure the balance response time, DDoS defense IP blocking was turned off and a scenario was assumed where user50 generates 50 threads with 100 loops of requests, setting the ramp-up to 0 so that only up to 50 requests can be made per second. User100 assumes a scenario where 100 threads with 100 loops of requests are generated, setting the ramp-up to 0 so that only up to 100 requests can be made per second. User200 assumes a scenario where 200 threads with 100 loops of requests are generated, setting the ramp-up to 0 so that only up to 200 requests can be made per second. User300 assumes a scenario where 300 threads with 100 loops of requests are generated, setting the ramp-up to 0 so that only up to 300 requests can be made per second. The summary report of the test is shown in Table 15. The transactions per second (TPS) were 560.73 without errors in 50 threads, 565.55 TPS with 0.01% errors in 100 threads, 482.35 TPS with 0.09% errors in 200 threads and 475.0 TPS with 0.7% errors in 300 threads, indicating an increase in errors as the number of threads increases.

**Table 15.** TC3 Response time of balance inquiry (without DDoS protection).

| Requests | Requests | | | | | | Response Times (ms) | | | | | Throughput | Network (KB/sec) | |
|---|---|---|---|---|---|---|---|---|---|---|---|---|---|---|
| Label | # of Samples | KO | Error% | Average | Min | Max | Median | 90th pct | 95th pct | 99th pct | | Transaction (Per sec) | Receive | Send |
| **Total** | **65,000** | **230** | **0.35%** | **425.91** | **0** | **1980** | **571.00** | **920.00** | **1022.00** | **1229.00** | | **383.81** | **168.20** | **82.17** |
| User50 | 5000 | 0 | 0.00% | 83.01 | 31 | 307 | 78.00 | 121.00 | 140.00 | 185.00 | | 560.73 | 242.54 | 120.47 |
| User100 | 10,000 | 1 | 0.01% | 156.17 | 15 | 638 | 136.00 | 276.00 | 329.00 | 426.00 | | 565.55 | 244.71 | 121.49 |
| User200 | 20,000 | 18 | 0.09% | 387.40 | 4 | 1496 | 368.00 | 651.00 | 737.00 | 896.99 | | 482.35 | 209.33 | 103.54 |
| User300 | 30,000 | 211 | 0.70% | 598.63 | 0 | 1980 | 571.00 | 920.00 | 1022.00 | 1229.00 | | 475.01 | 210.84 | 101.34 |

The results of processing with DDoS blocking enabled under the same conditions are shown in Table 16 and the DDoS defense logic is impacted by about 220 TPS in transactions per second: 313.39 for user50, 313.24 for user100, 232.1 for user200 and 223.06 for user300.

**Table 16.** TC4 response time of balance inquiry (with DDoS protection).

| Requests | Requests | | | | | | Response Times (ms) | | | | | Throughput | Network (KB/sec) | |
|---|---|---|---|---|---|---|---|---|---|---|---|---|---|---|
| Label | # of Samples | KO | Error% | Average | Min | Max | Median | 90th pct | 95th pct | 99th pct | | Transaction (Per sec) | Receive | Send |
| **Total** | **65,000** | **210** | **0.32%** | **830.81** | **0** | **3067** | **1131.00** | **1610.00** | **1764.00** | **2060.99** | | **223.60** | **97.89** | **47.88** |
| User50 | 5000 | 0 | 0.00% | 191.60 | 48 | 524 | 186.00 | 276.00 | 307.00 | 370.99 | | 247.34 | 106.98 | 53.14 |
| User100 | 10,000 | 1 | 0.01% | 372.60 | 8 | 1660 | 356.00 | 571.00 | 652.00 | 822.99 | | 252.31 | 109.17 | 54.20 |
| User200 | 20,000 | 32 | 0.16% | 752.27 | 3 | 2151 | 738.00 | 1112.90 | 1228.95 | 1468.00 | | 250.26 | 108.91 | 53.68 |
| User300 | 30,000 | 177 | 0.59% | 1142.44 | 0 | 3067 | 1131.00 | 1610.00 | 1764.00 | 2060.99 | | 251.95 | 111.39 | 53.81 |

In Figure 18a,b, the results are different when DDoS blocking is turned off and when DDoS blocking is not turned off, even though the conditions are the same. However, the difference between the two conditions is not significant, which shows that the performance of the proposed technique is excellent.

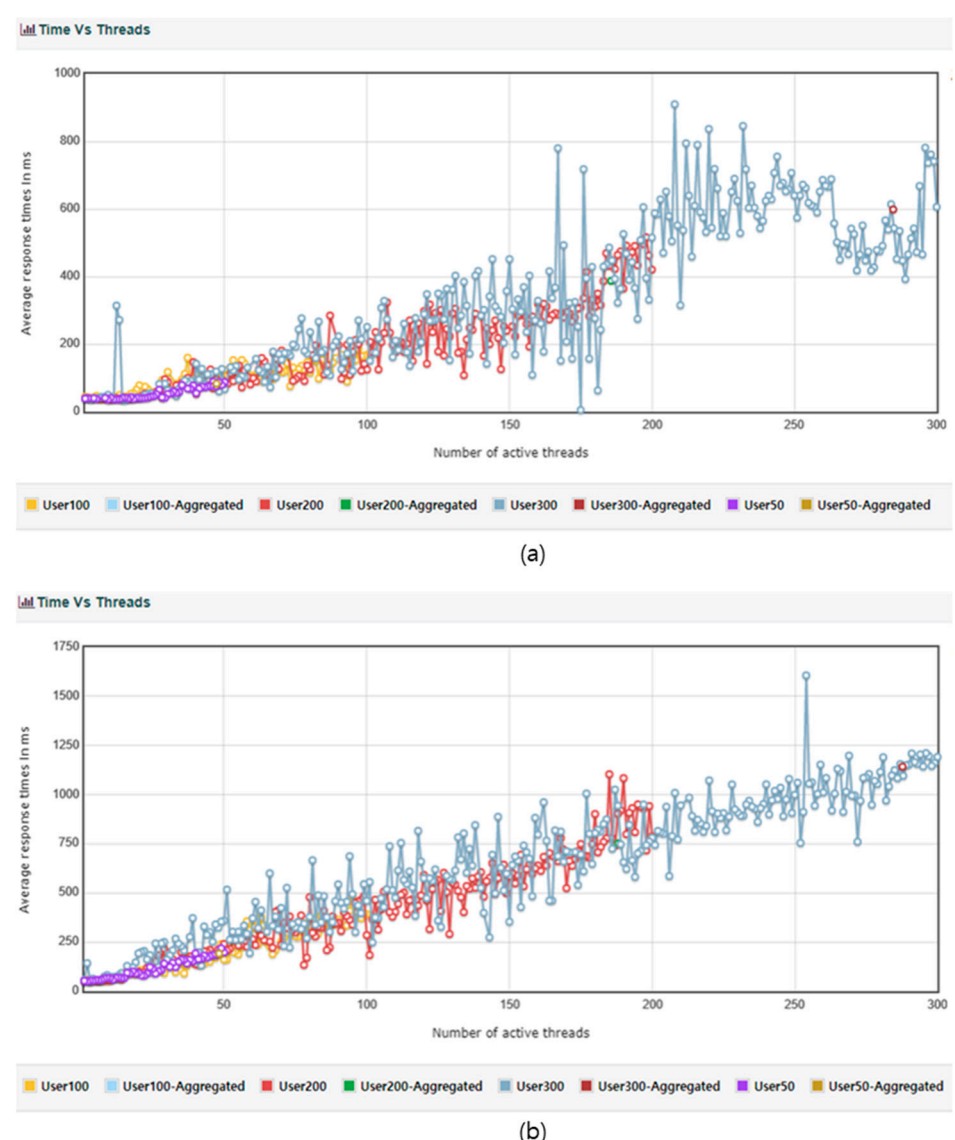

**Figure 18.** TC3 Active number of threads vs. response time: (**a**) without DDoS protection (**b**) with DDoS protection.

## 6. Conclusions

This paper has proposed a new server-based real-time P2P token payment blockchain system designed to address the problems of non-face-to-face transactions in the financial sector, especially in the situation where sophisticated voice phishing is spreading and reliance on open banking systems is increasing. This solution overcomes the limitations of private key management difficulties and gas fee management while utilizing the advantages of blockchain technology.

The server-based key management module effectively minimizes the risk of loss by securely storing private keys in keystore files and databases. The server-based token management module simplifies token generation and management, ensuring smooth token transactions through automatic gas charging. The transaction verification module, which utilizes transaction IDs without exposing personal information, guarantees transaction integrity and non-repudiation. In addition, advanced security measures such as external access IP blocking and DDoS protection are introduced to keep user data safe.

It also aims to provide convenient, secure and accessible online payment solutions to the general public by implementing identity verification using smartphones and platform-agnostic web applications. This comprehensive approach has the potential to innovate non-face-to-face transactions in the financial sector by providing user-friendly and secure alternatives to traditional methods. As the digital environment continues to evolve, the non-contact P2P real-time tokenized payment system can become an important tool for protecting users from financial fraud and creating a more inclusive and flexible financial ecosystem.

Through this research, it was demonstrated that blockchain token payments can be utilized as an anti-voice phishing technology by blocking the possibility of exposure to crime through

transparent transactions where all details can be made public and both the payer and the payee can be verified. Third-party authentication for non-repudiation is also possible. Furthermore, it is expected to create business opportunities for innovative finance, such as IoT automatic regular payments, small loan services and 24-h uninterrupted payment services. The feasibility and efficiency of the proposed system were confirmed through experimental results and it is expected to have a significant impact on the financial industry.

**Author Contributions:** Conceptualization, C.-S.J.; Methodology, C.-S.J.; Software, H.-J.K. and S.-S.H. All authors have read and agreed to the published version of the manuscript.

**Funding:** This research received no external funding.

**Institutional Review Board Statement:** Not applicable.

**Informed Consent Statement:** Not applicable.

**Data Availability Statement:** The data presented in this study are available on request form the corresponding author. The data are not publicly available due to research contract.

**Conflicts of Interest:** The authors declare no conflict of interest.

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
