# Peer review of "Non-Face-to-Face P2P (Peer-to-Peer) Real-Time Token Payment Blockchain System"

_applsci, doi:10.3390/app13137364_

Round 1
Reviewer 1 Report
The paper proposes a non-face-to-face P2P real-time token payment system that minimizes the risk of key loss by storing private keys in a keystore file and database through a server-based key management module. The authors argued that currently distributed ledgers have a limited adoption mainly due to the gas fees management as well as reinforces the open issues related to customer identity verification, transaction integrity, and transaction refusal. In contrast, the authors proposed system simplifies token creation and improved security through a server-based token management module and implements an automatic gas charging function for smooth token transactions.
The paper proposes a non-face-to-face P2P real-time token payment system that minimizes the risk of key loss by storing private keys in a keystore file and database through a server-based key management module.
The authors argued current distributed ledgers have a limited adoption mainly due to the gas fees management as well as reinforces the open issues related to customer identity verification, transaction integrity, and transaction refusal. In contrast, the authors proposed system simplifies token creation and improved security through a server-based token management module and implements an automatic gas charging function for smooth token transactions.
In the Introduction section the authors must reinforce the problem statement and paper contributions to better drive the audience through the main paper findings.
In the related works section, I would like to suggest removing subsections from 2.1-2.3 since they are too small. I still appreciate a better literature review of such technologies since they represent a hot topic in several research fields currently. Besides, based on the current works it is really hard to evaluate the real contribution of the current paper against the state-of-the-art.
Section 2.4 could be rewritten as Discussion and also should be improved to clarify the limitation mentioned as well.
I also recommend bringing section 2.5 out of the Related Work and rewriting it. This section must be improved in many directions from content to text presentation. For instance, in this section the authors need to better describe the limitations behind distributed ledgers, not focusing only on Ethereum. Maybe, the author could present a brief overview of other ledger technologies, compare them and summarize them in a table, and then compare with Ethereum.
Tables 1,2, and 3 must be better linked and explained. Otherwise, it is hard to understand the content and link the contribution.
Figure 6 must be replaced by a formula, and it should be explained.
Figure 7 needs to be better explained or suppressed since it represents a direct citation.
Figure 8 should be removed and its content explained in the text.
Section 3, I’d like to suggest removing the title of subsection 3.2 and move the text for the end of this section to improve paper proofreading.
Dear authors, the section 4 - Implementation is really hard to follow, it seems more a deployment rather than an implementation itself. It is not clear what was designed by your work and what descriptions belong to your implementation. Once the architecture was already presented in the section above! How the components interconnect with symverse/gsym, and which code was used to create the smart contract or any other security mechanism to sustain your proposed solution? In my opinion, this section needs to be rethought and re-designed to better express your work.
Some comments.
Use only English in Table 4, and please add a footnote for the link in subsection 4.2
Figure 16 is unreadable and could be remade or suppressed.
Figure 17 is a listing, not a figure.
Evaluation
Some comments:
Table 11,12,13,14 should be remade, and is unreadable.
Figure 25 and 26 should be placed side by side and the authors need to discuss why this difference happened, which is the implication for the current state-of-the-art and the benefits using the proposed work. It is the same comment for the rest of the evaluation.
Single “W” at line 463
It is not clear how authors may minimize the risk of loss by securely storing private keys in keystore files and databases. The evaluation could use more nodes to real simulate a blockchain scenario as well as the evaluation should happen under stress conditions to observe how the system behaves.
Reviewer 2 Report
The undoubted advantage of the work is the relevance of the describing problem and the detail of the presented research.
However, for approval it is necessary to emphasize what is the novelty of presented approach. Limitation of other solutions in section 2.4 does not contain a meaningful comparison. Authors have to answer some questions:
1. Why is this solution better than Ripple or AAVE, which are also focused on cross-chain financial p2p transfers?
2. The authors developed a set of services and use data integration with 1st level blockchains (Symverse). But there are confused phrase in section 4:
"The blockchain is integrated with the Symverse mainnet through a service node installed for the users' convenience"
3. Have the authors developed their own second-level blockchain? If it's true what parameters and what consensus algorithm does it use?
4. Is gas calculated for the network of the first level or the second?
5. The main result proposed by the authors is "The server-based key management module", but why is it safer than storing keys on costodial wallets, for example?
Round 2
Reviewer 1 Report
Dear authors, It is not possible to see how all of the points raised were addressed.
The text needs to be reviewed, and please address it carefully.
For instance:
* figure instead of Figure
* In Figure. 24 and In Figure. 24,
Many listings remain as figures (14, 15 e 16, maybe more?)
Figures from 19 to 22 could be grouped to improve proofreading.
The paper needs to be written from a scientific point of view. For instance, the related works section does not bring to the audience the relevance of your work, even less the state-of-the-art in such a context. You need to bring more paper for this sections and disccus over them.
It is not clear why did you remove your discussion when I asked to review it?
I also do not understand why the authors omitted the results with multiple peers. In my opinion, it represents the validation point of your work.
Att
Reviewer 2 Report
The authors have done a great job of finalizing their research for publication. Sections 2 and 5 now look more clear and logical.
Remaining issues:
1. It remains unclear how the authors simulate ddos attacks on one server.
2. Not taking into account other types of attacks, they declare a more reliable system than MetaMask.
3.The novelty of the approach lies in the fact that the method of storing several private keys is applied. If access to keystorage is obtained, then this approach does not make sense? Then it is not safer than Metamask.
Round 3
Reviewer 1 Report
Dear authors,
You keep pointing to tables and figures with lowercase letters instead of using uppercase letters, all the text needs to be verified following this verification.
See table/figure (not correct)
See Table/Figure (correct)
The Related Work section still needs revision.
Authors, are you sure no one else has implemented anything to solve the problem raised by your manuscript, or similar issues in the same direction? Please, add some more articles that discuss it. Otherwise, it is not possible to evaluate the impact of your work.
